# Investigating the association between early years foundation stage profile scores and subsequent diagnosis of an autism spectrum disorder: a retrospective study of linked healthcare and education data

Barry Wright [1], Mark Mon-Williams,[2] Brian Kelly,[3] Stefan Williams,[4] David Sims,[5] Faisal Mushtaq,[2] Kuldeep Sohal,[3] Jane Elizabeth Blackwell,[6] John Wright[3]

► Additional material is published online only. To view please visit the journal online (http://dx.doi.org/10.1136/bmjpo-2019-000483).

For numbered affiliations see end of article.

**Correspondence to**
Dr Barry Wright; barry.wright1@nhs.net

## ABSTRACT

**Objective** We set out to test whether the early years foundation stage profile (EYFSP) score derived from 17 items assessed by teachers at the end of reception school year had any association with autism spectrum disorder (ASD) diagnosis in subsequent years. This study tested the feasibility of successfully linking education and health data.

**Design** A retrospective data linkage study.

**Setting and participants** The Born in Bradford longitudinal cohort of 13 857 children.

**Outcome measures** We linked the EYFSP score at the end of reception year with subsequent diagnosis of an ASD, using all ASD general practitioner Read codes. We used the total EYFSP score and a subscore consisting of five key items in the EYFSP, prospectively identified using a panel of early years autism experts.

**Results** This study demonstrated the feasibility of linking education and health data using ASDs as an exemplar. A total of 8935 children had linked primary care and education data with 20.7% scoring <25 on the total EYFSP and 15.2% scoring <10 on a EYFSP subscore proposed by an expert panel prospectively. The rate of diagnosis of ASDs at follow-up was just under 1% (84 children), children scoring <25 on the total EYFSP had a 4.1% chance of ASD compared with 0.15% of the remaining children. Using the prospectively designed subscore, this difference was greater (6.4% and 0.12%, respectively).

**Conclusions** We demonstrate the feasibility of linking education and health data. Performance on teacher ratings taken universally in school reception class can flag children at risk of ASDs. Further research is warranted to explore the utility of EYFSP as an initial screening tool for ASD in early school years.

## INTRODUCTION

Large quantities of electronic data are generated routinely in health and education sectors. Linking these data could present an opportunity to improve information sharing

### What is known about the subject?

► Routine education and health data are rarely linked.
► To date, there is limited evidence that screening for autism spectrum disorders in the first 6 years of life is cost-effective.

### What this study adds?

► It is feasible to link routine education and health data for a cohort of children in England.
► Performance on educational measures taken universally in school reception class can flag children at risk of autism spectrum disorders.
► Linking these data has the potential to decrease costs associated with undiagnosed childhood conditions to the individual as well as health and education services.

and facilitate service user transitions within and across systems.

Ten per cent of children and young people aged 5–16 years have a clinically diagnosed mental disorder.[1] The short-term costs are estimated to be £1.58 billion per annum, with annual long-term costs calculated at £2.35 billion.[2] Neurodevelopmental impairments and conditions with high levels of need contributing to this cost have an estimated prevalence of around 3%–4% of children in England.[2] The Chief Medical Officer suggested that '*Commissioners and providers of services to children in primary education should develop and agree arrangements to ensure all primary schools adopt a comprehensive whole school approach to children's social and emotional wellbeing. They should provide specific help for*

## Box 1  Example Parental Quote

We requested a referral to paediatrician as we suspected autism when my son was 4 years. The special educational needs co-ordinator in his nursery had been involved when he was 3.5 years old as staff said he was not listening and his attention span was poor. We approached our health visitor after this to express our concerns. They referred him for speech and language therapy. The speech and language therapist referred him to the paediatrician. We waited ~2 years to be seen. After an initial appointment, we were told they wanted to wait and see. This meant that our son did not get appropriate treatment until he was 8 years of age. Should there not be a more systematic way of assessing children in need early?

*those children most at risk (or already showing signs) of social, emotional and behavioural problems'.*[2] The challenge facing policymakers and providers is how such a suggestion might be implemented. A useful starting point might arise from linking routine data across health and education services. ASD presents an exemplar condition that could elucidate the challenges and potential utility of data linkage across domains.

Autism spectrum disorders (ASDs) are neurodevelopmental disorders that lead to impaired reciprocal social interaction and repetitive or restricted patterns of behaviour[3] occurring in at least 1% of children in the UK.[4] The behavioural problems associated with the condition are a major cause of children being excluded from school.[2] A recent review showed that government policies and community resources impact on early identification of ASD[5] with evidence of geographic variation. Socioeconomic status and level of parental concern affect age of diagnosis.[5] Parents experience high levels of stress with the ASD diagnostic process, with over half dissatisfied with current UK services.[6] On average, families have to wait 3–4 years to receive a diagnosis.[6 7] Many children are identified early with a range of difficulties but not given a diagnosis of autism until much later.[7] Box 1 provides typical testimony from a parent consulted in our patient and public involvement work (slightly abridged).

Early intervention in ASD is associated with long-term symptom reduction.[8] This includes identifying appropriate educational placement early and parenting support interventions.[8] There has been a call for more sophisticated approaches to screening (such as stepped approaches or at-risk group identification) since whole population approaches have not proved cost-effective.[9]

A universal educational assessment is conducted on *all* children in their first year of schooling in the UK (the early years foundation stage profile; EYFSP). We predicted that scores on this developmental assessment might identify children at risk of neurodevelopmental problems. This could lead to earlier intervention to prevent the poor outcomes associated with undiagnosed conditions. Our prediction was motivated by the fact that information collected through the EYFSP routinely covers key domains of autism symptomatology, providing information about child's language development, social skills and emotional development.

## METHODS
### Design
This was a retrospective data linkage study. To test our hypothesis, we focused on ASD as an exemplar and retrospectively examined the EYFSP scores collected on the 13 857 children within the Born in Bradford (BiB) longitudinal birth cohort study.[10] This cohort has been followed up since birth. The BiB cohort comprises 12 453 women recruited at 28 weeks of pregnancy, who gave birth at the Bradford Royal Infirmary to 13 857 children between the period 2007 and 2011.[10] Half of all BiB families live within wards classed among the 20% most deprived within England and Wales. 45% of families are of Pakistani origin.[10]

### Patient and public involvement
A key strength of BiB has been its effective community engagement with considerable time and effort invested to ensure genuine coproduction of research. There is an active parent governor group and they advised on the protocol and materials used for this study. BiB has an active approach to partnership with families through regular newsletters, Facebook and Twitter with regular family network days and an annual family science festival.

Full informed consent was obtained for all participants. Cohort members gave their consent to access and link routine GP records via SystmOne, which currently has complete coverage of all GP practices in Bradford with secondary care records and education records. Figure 1 provides an overview of the recruitment process and the final sample included in the current study. Table 1 provides a comparison of the characteristics of the current sample with the whole BiB cohort.

### Outcome measures
#### EYFSP score
Table 2 displays the 17 items of the EYFSP, each scored 2 (meeting the level of development *expected*), 3 (*exceeding* this level) or 1 (not yet reaching this level—*emerging*). The items are designed to measure a range of educational, socioemotional, communicative and developmental factors. We used the total EYFSP score (the 'total score') and a subscore developed by a small group of ASD assessment experts prior to study (and so blind to the results) (a five-item 'subscore') (see box 2). The experts were academic and clinical child psychiatrists and psychologists with many years of ASD experience between them. The five items were chosen from the four main symptom areas defined in the WHO (1992)[3] research diagnostic criteria for ASD namely social reciprocity, language and communication, imagination delays and repetitive and stereotyped patterns of behaviour. Because the social reciprocity domain is given more weight in this classification system it was decided to include two items from the

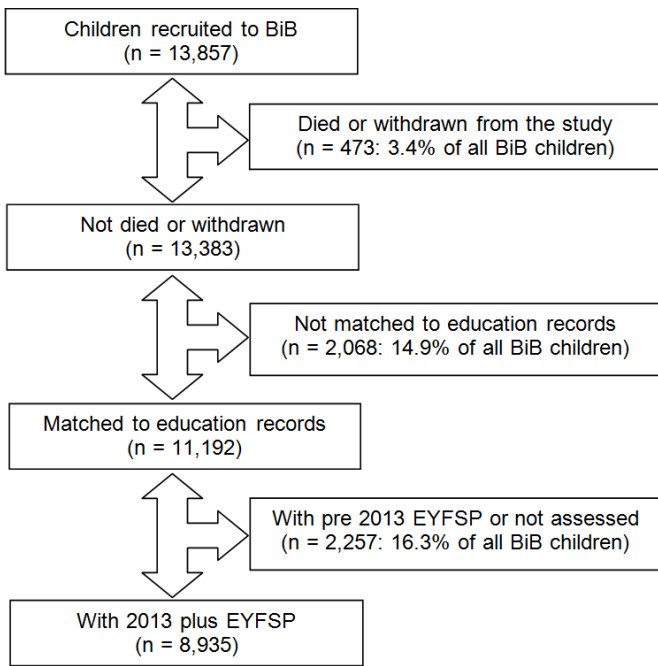

**Figure 1** Recruitment flowchart. BIB, Born in Bradford; EYFSP, early years foundation stage profile.

EYFSP to reflect this weighting. Children who underwent EYFSP assessment before 2013 were excluded from this study because the items measured in the EYFSP assessment were changed in 2013. The pre-2013 version is not compatible with the current EYFSP assessment.

### Read code diagnosis of ASD

The outcome measure for this study was the presence of a 'Read code' for an ASD recorded in a child's primary care records. A validated Read code list (see online supplementary material 2) has been shown in previous published work in ASD to be reliable and used with confidence to study ASD.[11]

| Table 1 Sample characteristics | | |
|---|---|---|
| | Sample in the current study (n=8,935) | Sample in the whole cohort (n=13,857) |
| **Gender** | | |
| Female | 49.1% | 49.0% |
| Male | 50.9% | 51.0% |
| **Ethnicity** | | |
| White British | 36.1% | 37.0% |
| Pakistani Heritage | 51.8% | 48.6% |
| Other | 12.1% | 14.4% |
| **Free school meal status** | | |
| In receipt | 24.7% | 25.1% |
| Not in receipt | 75.3% | 74.9% |

### Procedure

In collaboration with our primary care electronic health provider across Bradford, linkage was carried out using a complete deterministic match on National Health Service (NHS) number, surname, gender and date of birth (99% of BiB children matched to their health record) by the BiB data team. Routine electronic datasets include primary care data from the GP practice, hospital data, community care and education data. Hospital and maternity data were provided by Bradford Teaching Hospitals NHS Foundation Trust (BTHFT). The community care data were provided by Bradford District Care Trust. In collaboration with our local provider of electronic health records across primary care, the GP records were extracted by The Phoenix Partnership, SystmOne. The education dataset was provided by Bradford Metropolitan District Council. Our local authority, Bradford Metropolitan District Council linked BiB children to their Unique Pupil Identification Number (UPN) resulting in a match rate of 84% with their education records. The UPN is a 13-character code that identifies each pupil in the local-authority school system. As the local authority match on an iterative deterministic approach based on combinations of surname, date of birth, gender and postcode, if more than one match is made, then we do not receive any UPNs. At the point of analysis, 8935 children were included. Two thousand and sixty-eight children could not be linked mainly because the children were BiB but moved outside the area (~1200 children) or had moved in after EYFSP (~200) children. The other reason for exclusion was an inability to match to predetermined quality standards.

### Analysis

Logistic regression was employed to model the relationship between EYFSP score and the outcome of ASD diagnosis using Stata V.13.[12] A number of covariates were included in the analysis; gender (classified as male or female), ethnicity (White British, Pakistani Heritage and Other), free school meal status (whether in receipt of free school meals or not) and the age of the child (in years) at the date of GP data extract. Marginal effects were estimated[13] to produce predicted rates of ASD diagnosis for children based on whether they had a low EYFSP score.

The total EYFSP score ranged from 17 to 51, with a mean of 31.8 and a SD of 8.0. The five-item subscore ranged from 7 to 21, with a mean of 13.6 and a SD of 3.4 (see figure 2). However, the EYFSP scores are not normally distributed; mostly children score average on each item, but there are some who score higher and a fairly distinct group of children who have low scores resulting in a bimodal distribution of EYFSP scores, as indicated in figure 3. For this reason, the EYFSP scores are dichotomised for the purposes of analysis. Those scoring <25 in the total EYFSP score and <10 in the five-item subscore were categorised as low scores; these cut points equate to scores <1 SD below the mean (see figure 3).

**Table 2** Early years skills foundation profile: all items

| | |
|---|---|
| Physical development: health and self-care | EYFSP05 |
| Communication and language: listening and attention | EYFSP01 |
| Personal, social and emotional: managing feelings and behaviour | EYFSP07 |
| Communication and language: understanding | EYFSP02 |
| Expressive arts and design: being imaginative | EYFSP17 |
| Understanding the world: people and communities | EYFSP13 |
| Personal, social and emotional: self-confidence and self-awareness | EYFSP06 |
| Communication and language: speaking | EYFSP03 |
| Personal, social and emotional: making relationships | EYFSP08 |
| Expressive arts and design: exploring and using media and materials | EYFSP16 |
| Mathematics: shapes, space and measures | EYFSP12 |
| Physical development: moving and handling | EYFSP04 |
| Understanding the world: the world | EYFSP14 |
| Mathematics: numbers | EYFSP11 |
| Literacy: reading | EYFSP09 |
| Literacy: writing | EYFSP10 |
| Understanding the world: technology | EYFSP15 |

## RESULTS

Just under 1% (84 children) of the 8,935 children with matched general practitioner (GP) and education records had a diagnosis of autism. Of these, 1,852 (20.7%) children scored below the cut-off on the total EYFSP score, 72 of these children had an autism diagnosis. One thousand three hundred and fifty-five (15.2%) children scored below the cut-off on the EYFSP subscore, 72 of these children had an autism diagnosis (see table 3).

We found that children with a low EYFSP subscore were ~50 times more likely to have a diagnosis of autism compared with children without a low score. Males were twice as likely to have a diagnosis of ASD, while White British children and children not in receipt of free school meals had higher rates of diagnosis recorded in the GP records. Apart from age of the child, all differences were statistically significant and all covariates improved the model (see online supplementary material 1 for full model results). Analysis was carried out using the scores derived from all items and produced similar results; though the association was stronger for the EYFSP subscores (see figure 4).

Children with a low total EYFSP score (A) have a rate of autism of around 41 per 1,000 children (4.10%, 95% CI 3.08 to 5.13), far higher than children who do not have a low total EYFSP score (0.15%, 95% CI 0.05 to 0.25).

Children with a low five-item EYFSP subscore (B) have a rate of autism of around 64 per 1,000 children (6.38%, 95% CI 4.76 to 8.00), far higher than children who do not have a low five-item EYFSP subscore (0.12%, 95% CI 0.04 to 0.20).

---

### Box 2 Early years skills foundation profile (EYFSP): weighted subscore

Using a weighted subscore, where four aspects of childhood autism (social, language and communications, imagination and repetitive behaviour) were mapped onto EYFSP elements

**The social aspect mapped onto:**
► Personal, social and emotional: managing feelings and behaviour/.
► Personal, social and emotional: making relationships.

**Language and communications aspect mapped onto:**
► Communication and language: listening and attention.

**Imagination aspect mapped onto:**
► Expressive arts and design: being imaginative.

**Repetitive and stereotyped behaviours mapped onto:**
► Physical development: health and self-care.

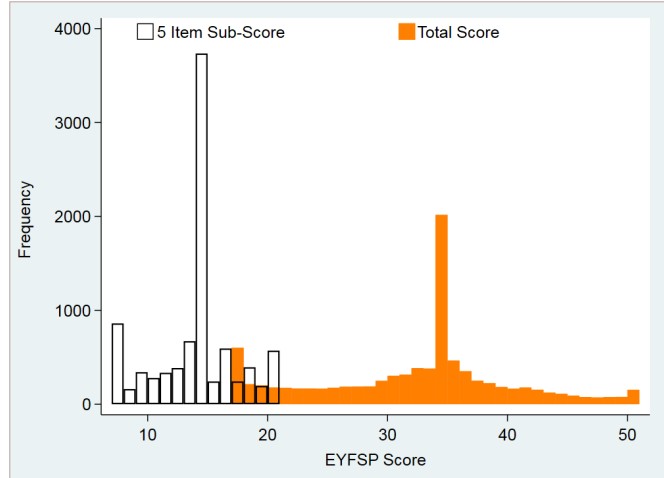

**Figure 2** Early years foundation stage profile (EYFSP): weighted subscore.

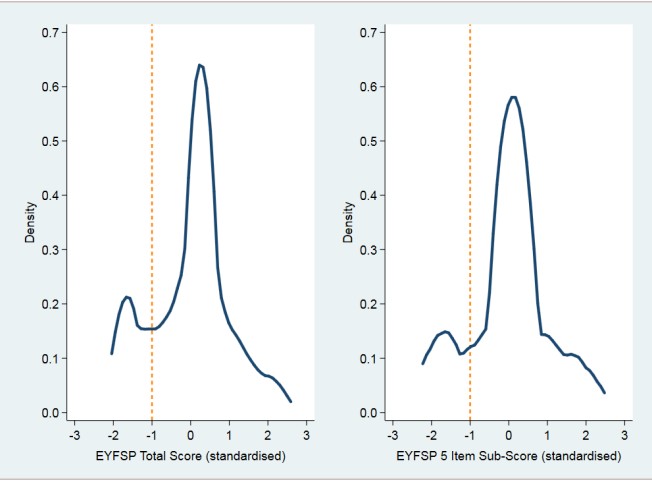

**Figure 3** Distribution of EYFSP scores—standardised with SD shown. EYFSP, early years foundation stage profile.

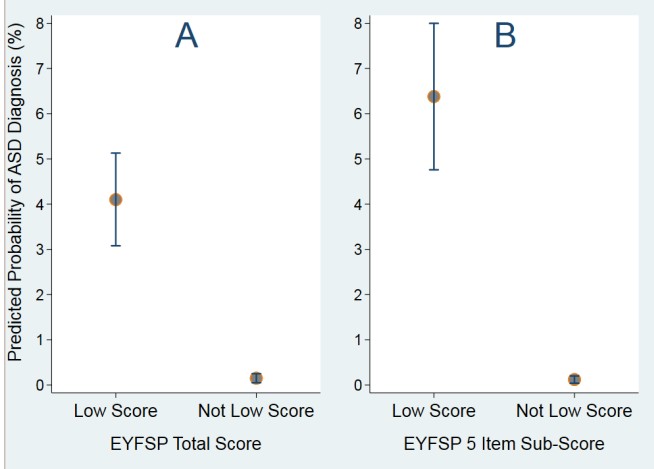

**Figure 4** Predicted rates of ASD for children by (A) total EYFSP score and (B) five-item EYFSP subscore (designed to map more closely to items that measure factors associated with autism). ASD, autismspectrum disorder; EYFSP,early years foundation stage profile.

## DISCUSSION

Our linkage of routine education and health data has revealed an association between school EYFSP score profile and the diagnosis of ASD. We focused this proof-of-concept study on ASD as: (1) the issues with late diagnosis are well documented; (2) early diagnosis is known to lead to enhanced parental interventions; and (3) improved educational pathways lead to better outcomes. The linkage process and subsequent analyses of the linked datasets has illustrated a potentially powerful way to create a smoother, more joined-up and timely early patient pathway to autism assessment by linking data across the currently tortuous pathway to referral for ASD assessment.

Gold standard assessment for ASD is lengthy and expensive[14] so care needs to be taken to ensure that screening has good specificity and sensitivity to avoid large numbers of unnecessary assessments. Further research is therefore necessary. Population screening for ASD is also expensive[14] and to date, the effectiveness of different screening strategies for ASD in reducing time to diagnosis has not been adequately tested.[9] While researchers have recommended that screening for ASD only be carried out in 'at risk' groups,[15] no localities or nations have yet successfully implemented or researched this. Our study suggests that a promising focus for future research lies in the use of routine educational data to identify children at risk. This could be highly cost-effective given the universal availability of this educational data.

Related school-based information is often collected as part of the assessment for neurodevelopment disorders (including ASD). Consequently, correlation between EYFSP scores and neurodevelopmental disorders may be expected. Routinely collected national school data may therefore be a cost-effective way of initially identifying children at risk; however, larger studies are required to adequately test this across neurodevelopmental disorders.

### Clinical and research implications

Encouraged by strong stakeholder support, we are now prospectively researching whether children identified by the EYFSP is a suitable at risk group for specific ASD screening and assessment for ASD in terms of acceptability, ethics and cost-effectiveness, and whether it is a sensitive and specific method of enabling prompt assessment leading to early intervention.

If successful it would substantially increase efficiencies in the delivery of current services. These include assessment resources and appropriate school placement/support earlier. It may also prevent multiple assessments from different professionals over time. We have recently shown an inequality of ASD diagnosis as a function of parental educational status[16] and the use of routine linked health and education data could be a powerful tool to help tackle such health inequalities.

Routine electronic data that are relevant to health and care pathways exist in fragments, spread across multiple organisations, each fragment with its own data controller. The challenges of cross-organisation coordination and information governance make it difficult to obtain linked data, and so most care providers and researchers have been unable to use it as a means of understanding whole-system

| Table 3 | Autism diagnosis and EYFSP scores | | |
|---|---|---|---|
| | **Low score** | **Not low score** | **Total** |
| **Total EYFSP score** | | | |
| With autism diagnosis | 72 | 12 | 84 |
| No autism diagnosis | 1780 | 7071 | 8851 |
| Total | 1852 | 7083 | 8935 |
| **EYFSP subscore** | | | |
| With autism diagnosis | 72 | 12 | 84 |
| No autism diagnosis | 1283 | 7568 | 8851 |
| Total | 1355 | 7580 | 8935 |

EYFSP, early years foundation stage profile.

healthcare. In Bradford, we have addressed this as part of The Connected Bradford Project, by reaching data sharing agreements with each of the health and social care organisations in the Bradford city region. Identifiable information is removed at source, and so personal information is not available to the project, which enables whole population data to be analysed. The pseudonym is derived from the NHS number, which crosses organisations, so that data can be linked after pseudonymisation. The Connected Bradford project at BTHFT applied to the Confidentiality Advisory Group (CAG) for approval under Regulation 5 of the Health Service (Control of Patient Information) Regulations 2002 to process confidential patient information without consent. Approved applications enable the data controller to provide specified information for the purposes of the relevant activity, without being in breach of the common law duty of confidentiality, although other relevant legislative provisions will still be applicable. Support was granted on the 3 September 2018 (CAG ref: 18/CAG/0091 and REC ref:18/YH/0200). BTHFT have permission under these regulations to provide personal details to the local authority to enable the linkage of education records for ~220 000 individuals of the Bradford population. There is good scope for this approach to be used across multiple trusts.

## CONCLUSIONS

Our 'proof-of-concept' study suggests that linking education and health data could improve the detection and support of children with neurodevelopmental problems such as ASD earlier—a priority area identified by the Chief Medical Officer.[2] The use of linked data to benefit outcomes is an important future goal.[17] In this context, our demonstration has implications beyond autism. It is becoming clear that the routine linkage of education and health data has the potential to drive efficiencies in children's services, facilitate early intervention and ultimately, improve quality of life for large numbers of children and their families.

**Author affiliations**
[1]Hull York Medical School and Dept Health Sciences, University of York, York, UK
[2]Institute of Psychological Sciences, University of Leeds, Leeds, UK
[3]Bradford Institute for Health Research, Bradford Teaching Hospitals NHS Foundation Trust, Bradford, UK
[4]Leeds Institute for Health Sciences, University of Leeds, Leeds, UK
[5]Child and Adolescent Mental Health Service, Bradford District Care NHS Foundation Trust, Saltaire, UK
[6]Child Oriented Mental Health Intervention Centre, Leeds and York Partnership NHS Foundation Trust, York, UK

**Correction notice** This article has been corrected since it was published. The license type of the paper has changed from CC BY-NC to CC BY.

**Acknowledgements** The Connected Bradford project is a Northern Health Science Alliance led programme funded by the Department of Health and delivered by a consortium of academic and NHS organisations across the north of England. The work uses data provided by patients and collected by the NHS as part of their care and support. We would like to acknowledge the support of Cathy Hulin and Poppy Konstantopoulou.

**Contributors** Contributed to the writing of the manuscript: BW, MM-W, BK, SW, DS, FM, KS, JEB and JW. Statistical analysis BK. Agree with the manuscript's results and conclusions: BW, MM-W, BK, SW, DS, FM, KS, JEB and JW. All authors have read, and confirm that they meet, ICMJE criteria for authorship.

**Funding** This work was supported by the UK Prevention Research Partnership (MR/S037527/1), which is funded by the British Heart Foundation, Cancer Research UK, Chief Scientist Office of the Scottish Government Health and Social Care Directorates, Engineering and Physical Sciences Research Council, Economic and Social Research Council, Health and Social Care Research and Development Division (Welsh Government), Medical Research Council, National Institute for Health Research, Natural Environment Research Council, Public Health Agency (Northern Ireland), The Health Foundation and Wellcome.

**Disclaimer** The views expressed are those of the author(s) and not necessarily those of the NHSA, NHS or the Department of Health.

**Competing interests** None declared.

**Patient consent for publication** Not required.

**Ethics approval** Ethical approval was granted by Bradford NHS Research Ethics Committee (Ref 07/H1302/112).

**Provenance and peer review** Not commissioned; externally peer reviewed.

**Data availability statement** Data are available upon reasonable request.

**ORCID iD**
Barry Wright http://orcid.org/0000-0002-8692-6001

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
