## [Reviewer comments · BMJ Paediatrics Open]

This paper was submitted to a another journal from Archives of Disease in Childhood but declined for publication following peer review. The authors addressed the reviewers' comments and submitted the revised paper to BMJ Paediatrics Open. The paper was subsequently accepted for publication at BMJ Paediatrics Open.

ARTICLE DETAILS

TITLE (PROVISIONAL)	Investigating the association between early years foundation stage profile scores and subsequent diagnosis of an autism spectrum disorder: A retrospective study of linked healthcare and education data
AUTHORS	Wright, Barry; Mon-Williams, Mark; Kelly, Brian; Williams, Stefan; Sims, David; Mushtaq, Faisal; Sohal, Kuldeep; Blackwell, Jane; Wright, Prof John

VERSION 1 – REVIEW

REVIEWER	Reviewer name: Harron, Katie Institution and Country: UCL Great Ormond Street Institute of Child Health Competing interests: no competing interests
REVIEW RETURNED	14-Oct-2018

GENERAL COMMENTS	The researchers have achieved something valuable in this study and I think there are important implications for future work in linking education and health data. However, I don't feel that this paper does the study justice. It attempts to both test the feasibility of successful linkage, and also to test whether EYFSP scores predict Autism Spectrum Disorder. However, I think neither aim is given the required detail within this manuscript. The comments I have included below are aimed at improving the usefulness of this paper for other researchers. Re the first aim of testing whether EYFSP is associated with ASD in subsequent years, I understand that this is just one possible exemplar and the aim of the study may not have been to develop a validated test. However I think the methodological details should still be provided. 1. Table 1 is not referred to within the text.2. More detail should be given on how the 5 item sub-score was derived and how consensus was achieved by the assessment experts. Was this designed in a systematic way or might other experts come up with another sub-score? How exactly was the sub-score weighted?3. More detail should be given on the validity of the Read codes for ASD. Were any attempts made to find cases of ASD that were not captured by the code list? i.e. how was the sensitivity and specificity of the code list assessed?
---

It is not clear how the code list was derived, since the manuscript states that 'post-hoc' examination revealed which read codes were recorded. The Read code list should be provided (in an appendix if necessary).

4. Table 3 is actually a figure.

5. It would be helpful to give percentages in Table 4. However I'm not sure of the usefulness of this table, given we aren't given details of how to derive the weighted sub-score. This information could perhaps be more easily given by an extra line (the cut-off) on the graph in table 3.

6. How was the cut-off for high and low sub-scores decided upon, i.e. why were scores below 10 considered as low? Was a ROC curve used, or any other methods of selecting an optimal cut-off? Similarly for the total score.

7. Very little information on the model building is provided. How were the covariates (e.g. ethnicity / age) categorized? Were all of these covariates significantly associated with the outcome? All model coefficients from the logistic regression should be provided, along with confidence intervals. The authors should comment on the usefulness of including covariates in the model. If such a model were to be used in practice to predict future outcomes, would these covariates be routinely available? Would the addition of these covariates substantially improve on a model with EYFSP only?

8. Finally, consideration should be given to the reproducibility of these results. If this score were to be used in practice, it would need to be validated. This could have been done within this study, e.g. by using development / test data samples, or through the use of bootstrapping.

9. The lines on Figure 1 should be labeled (confidence intervals?)

Considering the second aim, i.e. feasibility of linkage, I think again that additional details would be really helpful for other researchers.

10. More detail on the consent model should be provided. Were cohort members originally consented for linkage, or was this done retrospectively? Did any participants not consent to linkage or to accessing GP records? What impact might this have had on the sample?

11. More information on the linkage procedure would be helpful. Who carried out the linkage? Was this done using a set of deterministic rules, or exact matches only, or probabilistic linkage?

12. The text states that linkage was carried out for 'education records'. Does this actually mean cohort records that contained information on education? The authors should discuss whether such linkage would also be feasible on a national level – this is important for the conclusions / implications of this study.

13. Why were EYFSP results only available for 9029 of the original cohort? A consort-type Figure 1 would be helpful to demonstrate the success of the linkage and any missing data.

	14. Do the authors have insight into why some records could not be linked? Were the unlinked records random or a particular subset? Were any attempts made to identify any falsely-linked records? It may be that this evaluation was not possible due to the pseudonymisation at source – but if this is the case, the implications for quality assessment should be carefully discussed.
--	---

REVIEWER	Reviewer name: Bellman, Martin Institution and Country: Royal Free Hospital, RNTNEH Competing interests: none
REVIEW RETURNED	01-Nov-2018

GENERAL COMMENTS	The methodology of this study is interesting as it is based on linkage between computer datasets of two different agencies - health and education. The authors demonstrate that it is feasible but it would be helpful to be reassured that the data are valid from each source and that the individuals identified are in fact the same person. The diagnosis of ASD relies on the accuracy of the primary care dataset (Read codes) and the reliability of that may be deficient - please could the authors comment on the criteria used for making the diagnosis and the reliability of the process for entering it into the computer system. Similar questions apply to the teachers' assessment for the EYFSP. The subjects are from the Born in Bradford study, which the authors point out is a very deprived population. How many of the children were not registered or included on a primary care register and/or were not in a school where an EYFSP assessment was done? I am unclear about how representative the study cohort is of the population as the BiB study included 13,500 children and matched linkage of data was done on 8854 (although the number in Table 4 is 9013) - please could the authors explain that, perhaps with a flow chart, and comment on any differences between included and not-included children. The statement "Early intervention in ASD is crucial for good outcomes" is a little controversial and needs some explanation. The authors say correctly that screening for ASD is difficult and has not been implemented anywhere. How does the study impact on that conclusion? It is not surprising that there is a strong association between a low EYFSP score and the diagnosis of ASD as school information is an essential part of the diagnostic process and therefore the two factors are not independent. ASD is cited as an 'exemplar' for data linkage and maybe that needs some expansion as the reported study is specifically about EYFSP scores and ASD diagnosis and it is not explained how this allows the generalisation that data linkage can be a powerful tool to "improve the detection and support of children with neurodevelopmental problems" and "improve quality of life for large numbers of children and their families".
--

VERSION 1 – AUTHOR RESPONSE

Reviewer: 1

Comments to the Author

1. Re the first aim of testing whether EYFSP is associated with ASD in subsequent years, I understand that this is just one possible exemplar and the aim of the study may not have been to develop a validated test. However I think the methodological details should still be provided.

We have provided considerably more methodological detail including additional information about:

- Recruitment and consent (please see pages 5 & 6 and Figure 1: Recruitment flow chart)
- Data linkage (please see page 11)
- The outcome measures (please see pages 7 and 8 and supplementary material 3).
- The analysis (please see page 11 and supplementary material 1).

All page numbers mentioned refer to the final version of the paper (without track changes).

2. Table 1 is not referred to within the text.

The table “Early Years Skills Foundation Profile: All items” was originally Table 1 and is now Table 2 due to another table being added. This table is mentioned in the text:

“Table 2 displays the 17 items of The Early Years Foundation Stage Profile, each scored 2 (meeting the level of development expected), 3 (exceeding this level) or 1 (not yet reaching this level - emerging). The items are designed to measure a range of educational, socio-emotional, communicative and developmental factors.”

Please see page 7.

3. More detail should be given on how the 5 item sub-score was derived and how consensus was achieved by the assessment experts. Was this designed in a systematic way or might other experts come up with another sub-score? How exactly was the sub-score weighted?

We have explained how the five item sub-score was derived and how the sub-score was weighted:

“We used the total EYFSP score (the ‘total score’) and a sub-score developed by a small group of ASD assessment experts prior to study and blind to the results (a 5 item ‘sub-score’) (see table 3). The experts were academic and clinical child psychiatrists and psychologists with many years of ASD experience between them. The five items were chosen from the four main symptom areas defined in the World Health Organisation (1992) [3] research diagnostic criteria for ASD namely social reciprocity, language and communication, imagination delays and repetitive and stereotyped patterns of behaviour. Because the social reciprocity domain is given more weight in this classification system it was decided to include two items from the EYFSP to reflect this weighting. Sensitivity analysis was carried out using the scores derived from all items.”

Please see pages 7 and 8.

4. More detail should be given on the validity of the Read codes for ASD. Were any attempts made to find cases of ASD that were not captured by the code list? i.e. how was the sensitivity and specificity of the code list assessed? It is not clear how the code list was derived, since the manuscript states that ‘post-hoc’ examination revealed which read codes were recorded. The Read code list should be provided (in an appendix if necessary).

The Read code list has been provided in the supplementary files (please see supplementary file 3). The validated Read code list has been shown in previous published work in ASD to be reliable and used with confidence to study ASD [11].

“Read Code Diagnosis of ASD

The outcome measure for this study was the presence of a ‘Read code’ for an ASD recorded in a child’s primary care records. A validated Read code list (see supplementary material 3) has been shown in previous published work in ASD to be reliable and used with confidence to study ASD [11].”

Please see page 8.

In the current study children are classified as having a diagnosis of ASD in cases where the Read code is recorded in their GP records. Unfortunately it was not possible to test the sensitivity and specificity of this diagnosis as part of this retrospective data linkage study. However, given the difficulty in obtaining a diagnosis of ASD in childhood, it is unlikely that there will be any children with a diagnosis of ASD who do not have ASD. As the read codes are used to record a diagnosis rather than a diagnostic test, establishing the sensitivity and specificity of Read codes is not appropriate. There are no children with a diagnosis of ASD who are not captured by the Read codes. However, there may be a number of children with ASD who have not yet received a diagnosis. Unfortunately we do not have this information.

5. Table 3 is actually a figure.

Thank you for highlighting this. The title has been changed to “Figure 2: Early Years Foundation Stage Profile (EYFSP): Weighted sub-score.”

Please see the figure legends at the end of the document.

6. It would be helpful to give percentages in Table 4. However I’m not sure of the usefulness of this table, given we aren’t given details of how to derive the weighted sub-score. This information could perhaps be more easily given by an extra line (the cut-off) on the graph in table 3.

As suggested, this table has been removed.

7. How was the cut-off for high and low sub-scores decided upon, i.e. why were scores below 10 considered as low? Was a ROC curve used, or any other methods of selecting an optimal cut-off? Similarly for the total score.

We have explained how the cut off for sub-scores and total scores was designed in this study and added the text accordingly:

“The total EYFSP score ranged from 17 to 51, with a mean of 31.8 and a standard deviation of 8.0. The five item sub-score ranged from 7 to 21, with a mean of 13.6 and a standard deviation of 3.4 (see figure 2). However, the EYFSP scores are not normally distributed; mostly children score average on each item, but there are some who score higher and a fairly distinct group of children who have low scores resulting in a bimodal distribution of EYFSP scores, as indicated in figure 3. For this reason the EYFSP scores are dichotomised for the purposes of analysis. Those scoring below 25 in the total EYFSP score and below 10 in the five item sub-score were categorised as low scores; these cut points equate to scores less than one standard deviation below the mean (see figure 3).”

Please see page 11.

8. Very little information on the model building is provided. How were the covariates (e.g. ethnicity / age) categorized? Were all of these covariates significantly associated with the outcome? All model coefficients from the logistic regression should be provided, along with confidence intervals.

The authors should comment on the usefulness of including covariates in the model. If such a model were to be used in practice to predict future outcomes, would these covariates be routinely available? Would the addition of these covariates substantially improve on a model with EYFSP only?

We have added further information about the model building and the covariates. The results of the logistic regression models are displayed in supplementary material 1.

“Logistic regression was employed to model the relationship between EYFSP score and the outcome of ASD diagnosis using Stata 13 [12]. A number of covariates were included in the analysis; gender (classified as male or female), ethnicity (White British, Pakistani Heritage and Other), free school meal status (whether in receipt of free school meals or not) and the age of the child (in years) at the date of GP data extract. Marginal effects were estimated [13] to produce predicted rates of ASD diagnosis for children based on whether they had a low EYFSP score.”

Please see page 11.

9. Finally, consideration should be given to the reproducibility of these results. If this score were to be used in practice, it would need to be validated. This could have been done within this study, e.g. by using development / test data samples, or through the use of bootstrapping.

The plan is to test the reproducibility of these results in a new study which is already underway. This involves prospectively using the EYFSP and assessing all children below the threshold and a random sub-sample of those above the threshold.

10. The lines on Figure 1 should be labeled (confidence intervals?)

The Early Years Foundation Stage Profile (EYFSP): Weighted sub-score Figure (now Figure 2 due to an additional figure being added) has been correctly labelled.

11. More detail on the consent model should be provided. Were cohort members originally consented for linkage, or was this done retrospectively? Did any participants not consent to linkage or to accessing GP records? What impact might this have had on the sample?

Full informed consent was obtained for all participants. Ethical approval was granted by Bradford NHS Research Ethics Committee (Ref 07/H1302/112). Cohort members gave their consent at recruitment for the study team to access their medical records. GP records were extracted from SystmOne, which currently has complete coverage of all GP practices in Bradford.

12. More information on the linkage procedure would be helpful. Who carried out the linkage? Was this done using a set of deterministic rules, or exact matches only, or probabilistic linkage?

GP records were extracted where there was a complete deterministic match on NHS number, surname, date of birth and gender.

Education records were linked in an iterative deterministic matching approach based on different combinations of surname, date of birth, gender and postcode.

13. The text states that linkage was carried out for ‘education records’. Does this actually mean cohort records that contained information on education? The authors should discuss whether such linkage would also be feasible on a national level – this is important for the conclusions / implications of this study.

GP records were extracted where there was a complete deterministic match on NHS number, surname, date of birth and gender.

Education records were linked in an iterative deterministic matching approach based on different combinations of surname, date of birth, gender and postcode.

Our 'proof-of-concept' study suggests that linking education and health data could improve the detection and support of children with neurodevelopmental problems such as ASD earlier – a priority area identified by the Chief Medical Officer [2]. The use of linked data to benefit outcomes is an important future goal [17]. In this context our demonstration has implications beyond autism. It is becoming clear that the routine linkage of education and health data has the potential to drive efficiencies in children's services, facilitate early intervention and ultimately, improve quality of life for large numbers of children and their families

14. Why were EYFSP results only available for 9029 of the original cohort? A consort-type Figure 1 would be helpful to demonstrate the success of the linkage and any missing data.

Figure 1 now displays recruitment and data linkage information.

15. Do the authors have insight into why some records could not be linked? Were the unlinked records random or a particular subset? Were any attempts made to identify any falsely-linked records? It may be that this evaluation was not possible due to the pseudonymisation at source – but if this is the case, the implications for quality assessment should be carefully discussed.

Not all children were linked to education data.

Reviewer: 2

Comments to the Author

1. The methodology of this study is interesting as it is based on linkage between computer datasets of two different agencies - health and education. The authors demonstrate that it is feasible but it would be helpful to be reassured that the data are valid from each source and that the individuals identified are in fact the same person.

Linkage was carried out using NHS number, surname and date of birth (99% of children matched) and education records. The final sample included 8, 935 children.

We have given provided considerably more methodological detail including additional information about:

- Recruitment and consent (please see pages 5 & 6 and Figure 1: Recruitment flow chart)
- Data linkage (please see page 11)
- The outcome measures (please see pages 7 and 8 and supplementary material 3).
- The analysis (please see page 11 and supplementary material 1).

2. The diagnosis of ASD relies on the accuracy of the primary care dataset (Read codes) and the reliability of that may be deficient - please could the authors comment on the criteria used for making the diagnosis and the reliability of the process for entering it into the computer system. Similar questions apply to the teachers' assessment for the EYFSP.

A validated Read code list (please see supplementary material 3) has been shown in previous published work in ASD to be reliable and used with confidence to study ASD [11].

Read codes are used by general practitioners for recording confirmed diagnosis from clinical services. We are now carrying out a pro-respective study where we are comprehensively assessing children with low EYFSP scores with an autism assessment and a random sub-sample of those above the threshold, we hope to publish this at the end of 2019.

3. The subjects are from the Born in Bradford study, which the authors point out is a very deprived population. How many of the children were not registered or included on a primary care register and/or were not in a school where an EYFSP assessment was done? I am unclear about how representative the study cohort is of the population as the BiB study included 13,500 children and matched linkage of data was done on 8854 (although the number in Table 4 is 9013) - please could the authors explain that, perhaps with a flow chart, and comment on any differences between included and not-included children.

Figure 1 has been added to display the recruitment and data linkage data.

4. The statement "Early intervention in ASD is crucial for good outcomes" is a little controversial and needs some explanation.

We have altered this statement to now read: "Early intervention in ASD is associated with long-term symptom reduction [8]. This includes identifying appropriate educational placement early and parenting support interventions [8]. There has been a call for more sophisticated approaches to screening (such as stepped approaches or at risk group identification) since whole population approaches have not proved cost-effective [9]".

Further information about an early intervention trial in ASD:

Pickles et al. (2016) conducted a randomised controlled trial of a parent-mediated social communication intervention (PACT) for children aged 2-4 years with core autism. The children were followed up approximately 6 years (median= 5.75 years) after the trial endpoint. The results showed long-term symptom reduction (reduction in restricted and repetitive behaviours, improved social communication) following early intervention in autism spectrum disorder. https://ac.els-cdn.com/S0140673616312296/1-s2.0-S0140673616312296-main.pdf?_tid=20d9a0c3-c21c-4734-86b7-ada82bbc2f17&acdnat=1548771612_e8d418aad1d5790f8d37e794a01b2d1e

5. The authors say correctly that screening for ASD is difficult and has not been implemented anywhere. How does the study impact on that conclusion?

Our 'proof-of-concept' study suggests that linking education and health data could improve the detection and support of children with neurodevelopmental problems such as ASD earlier – a priority area identified by the Chief Medical Officer [2]. The use of linked data to benefit outcomes is an important future goal [17]. In this context our demonstration has implications beyond autism. It is becoming clear that the routine linkage of education and health data has the potential to drive efficiencies in children's services, facilitate early intervention and ultimately, improve quality of life for large numbers of children and their families.

6. It is not surprising that there is a strong association between a low EYFSP score and the diagnosis of ASD as school information is an essential part of the diagnostic process and therefore the two factors are not independent.

We agree. We have given some explanation of this in the discussion.

"Related school based information is often collected as part of the assessment for neurodevelopment disorders (including ASD).

Consequently correlation between EYFSP scores and neurodevelopmental disorders may be expected. Routinely collected national school data may therefore be a cost effective way of initially identifying children at risk, however larger studies are required to adequately test this across neurodevelopmental disorders”.

7. ASD is cited as an 'exemplar' for data linkage and maybe that needs some expansion as the reported study is specifically about EYFSP scores and ASD diagnosis and it is not explained how this allows the generalisation that data linkage can be a powerful tool to "improve the detection and support of children with neurodevelopmental problems" and "improve quality of life for large numbers of children and their families".

Information has been added to clarify this.

“Related school based information is often collected as part of the assessment for neurodevelopment disorders (including ASD). Consequently correlation between EYFSP scores and neurodevelopmental disorders may be expected. Routinely collected national school data may therefore be a cost effective way of initially identifying children at risk, however larger studies are required to adequately test this across neurodevelopmental disorders”.

VERSION 2 – REVIEW

REVIEWER	Reviewer name: Katie Harron Institution and Country: UCL, UK Competing interests: no competing interests
REVIEW RETURNED	29-Apr-2019

GENERAL COMMENTS	This is a really nice paper and an important study, but there are some inconsistencies throughout the manuscript. I have a few comments aiming to improve the presentation of results. The text is lacking in details about how the datasets were obtained and linked – this information would be useful for other researchers, who might want to build on the experience of BiB. Since you are aiming to demonstrate the feasibility of linking education and health data, it would be helpful to provide the number of participants that were linked in the abstract. Please also provide the numbers (%) scoring below each of the cut offs. What this study adds – the first bullet point should be more specific, e.g. for a cohort of children in England. The inclusion of the parent testimony is really valuable and I commend the authors for including this. Figure 1 is helpful – please add percentages for each number of excluded children. Why were children with pre 2013 EYFSP excluded? Is there any information on why 2068 children could not be linked, e.g. was this mainly due to missing or incorrect identifiers, or could it have been that their record wasn't available in the education data? The main text states that 99% of children were linked – but $2068/13383 = 15\%$. Please clarify. There needs to be a section on the datasets that were linked.
--

	Where did the education dataset come from? How were the GP records extracted? Who performed the linkage? The procedure for linking says that NHS number was used – but NHS number is not routinely recorded on education records. Please clarify this process. I'm a little confused about the scores that were used. The methods starts by saying that the total score was used, and then that a subscore was used. I assume that the total score was the primary outcome. Then a sensitivity analysis is mentioned – I'm not clear what the difference between this and the total score is. Then the results focus on the subscore, which now seems to be the primary outcome. This all needs to be made clear and consistent throughout the methods and results. The text states that cohort members gave their consent to access GP records – but it does not mention education records. Was explicit consent obtained for linkage with education records? The results section is rather brief. First sentence – please give actual numbers. Start with the full score (keep the same order as the methods and abstract) and give how many children scored above / below each cut off. Also give the numbers and percentages in each category with a diagnosis, and for each subgroup that you mention in the text. It would be helpful to include these characteristics in a table in the main text, along with the model coefficients that are currently in the supplementary material. This could be in the main text for the primary analysis (total score) and left in the appendix for the subscore. For the model coefficients, please provide actual p-values rather than just * <0.05 (or leave them out altogether). Supplementary material 2 seems to repeat Figure 3 from the main text. Rather than the current Figure 4, which just presents four probabilities, it would be more helpful to present the distribution of scores for children with and without diagnoses, e.g. using histograms. This is helpful as it would show to what extent scores for the two groups of children overlap. Please define UPN.
--	---

REVIEWER	Reviewer name: Jane Waite Institution and Country: Aston University Competing interests: None
REVIEW RETURNED	11-Aug-2019

GENERAL COMMENTS	Thank you for this opportunity to review this revised manuscript for publication. I have read the entire manuscript and the author's response to the reviewers' comments. It appears that the authors have made all of the changes that have been requested during the first round of review and that they have provided sufficient detail. I have some reservations about the results as it is not particularly surprising that children with lower EYFSP scores are those that are more likely to have an autism diagnosis, given intellectual disability is associated with autism. I am also unsure about the authors' decision to map the 'physical development - health and self-care'
---

	code to restricted repetitive behaviours as these do not seem to be measuring the same thing. Restricted repetitive behaviours are far more specific than physical development, and physical development is likely picking up on degree of disability rather than autism. Despite this, I understand that the purpose of this study is to demonstrate the utility of linking datasets and identifying at risk groups rather than being a specific diagnostic tool, and therefore I believe the study meets these aims.
--	--

VERSION 2 – AUTHOR RESPONSE

Reviewer 1

Comment: The text is lacking in details about how the datasets were obtained and linked – this information would be useful for other researchers, who might want to build on the experience of BiB.

Response: Thank you for asking for more detail. We have included the following information in the paper (please see the ‘procedure’ section on page 11):

Routine electronic datasets include primary care data from the GP practice, hospital data, community care and education data. Hospital and maternity data was provided by Bradford Teaching Hospitals NHS Foundation Trust.

The community care data was provided by Bradford District Care Trust. In collaboration with our local provider of electronic health records across primary care, the GP records were extracted by The Phoenix Partnership, SystemOne. The education dataset was provided by Bradford Metropolitan District Council.

The healthcare and education data linkage was completed by the BiB Data Team using a variety of patient confidential information.

Comment: Since you are aiming to demonstrate the feasibility of linking education and health data, it would be helpful to provide the number of participants that were linked in the abstract. Please also provide the numbers (%) scoring below each of the cut offs.

Response: We have now added this detail as follows:

8,935 children were linked

1,852 (20.7%) children scored below the cut off on the total EYFSP score.

1,355 (15.2%) children scored below the cut off on the EYFSP sub score.

This has been added to the text in the abstract.

Comment: What this study adds – the first bullet point should be more specific, e.g. for a cohort of children in England.

Response: We have added this information to the paper. Please see the “what this study adds” section.

Comment: The inclusion of the parent testimony is really valuable and I commend the authors for including this.

Response: Many thanks for this comment.

Comment: Figure 1 is helpful – please add percentages for each number of excluded children.

Response: Figure 1 has been amended with the percentage of children excluded at each stage added.

Comment: Why were children with pre 2013 EYFSP excluded?

Response: The items measured in the EYFSP assessment were changed in 2013; the pre 2013 version is not compatible with the current EYFSP assessment. We wanted to use an instrument currently in use.

We have added this information to the text (please see page 8 under the section 'EYFSP score').

Comment: Is there any information on why 2068 children could not be linked, e.g. was this mainly due to missing or incorrect identifiers, or could it have been that their record wasn't available in the education data?

Response: Education records were linked in an iterative deterministic matching approach based on different combinations of surname, date of birth, gender and postcode.

The Bradford LA education team match on these combinations of patient confidential information and provide the BiB data team a list of the Unique Pupil Identification Numbers (UPNs). If more than one match is made, then no UPN is given and they are excluded.

As shown in figure 1 there were 2,068 (around 15% of the BiB children) who were not matched to education records. Education records were only available for children attending schools in the local authority area. Most of these cases relate to children who were born in Bradford but moved to another local authority prior to the EYFSP assessment (around 1,200 children) or who had been born outside the Bradford local authority area (around 200 children).

We have added the following text to the procedure section (page 11):

2068 children could not be linked mainly because the children were born in Bradford but moved outside the area (approximately 1,200 children) or had moved in after EYFSP (approximately 200). The other reason for exclusion was an inability to match to pre-determined quality standards.

Comment: The main text states that 99% of children were linked – but $2068/13383 = 15\%$. Please clarify.

Response: Thank you-we have now clarified this. We now include additional details on this process in the procedure section of the manuscript (please see page 11).

GP records were extracted where there was a complete deterministic match on NHS number, surname, date of birth and gender. Hence 99% of the BiB cohort were linked.

Education records were linked in an iterative deterministic matching approach based on different combinations of surname, date of birth, gender and postcode.

Essentially 99% of the BiB cohort was matched to GP records, but around 15% did not have linkage to education records. This relates to the comments added to the above (in point 7) on why the linkage to education records was less than to GP records.

Comment: There needs to be a section on the datasets that were linked. Where did the education dataset come from? How were the GP records extracted? Who performed the linkage?

Response: We have clarified this in the text (please see the 'procedure' section on page 11.)

Routine electronic datasets include primary care data from the GP practice, hospital data, community care and education data. Hospital and maternity data was provided by Bradford Teaching Hospitals NHS Foundation Trust.

The community care data was provided by Bradford District Care Trust. In collaboration with our local provider of electronic health records across primary care, the GP records were extracted by The Phoenix Partnership, SystemOne. The education dataset was provided by Bradford Metropolitan District Council.

The healthcare and education data linkage was completed by the BiB Data Team using a variety of patient confidential information.

Comment: The procedure for linking says that NHS number was used – but NHS number is not routinely recorded on education records. Please clarify this process.

Response: Healthcare data linkage uses a combination of patient confidential information including the NHS number.

To link healthcare and education data a combination of patient confidential information is used.

Please see the answer to items 1, 7 and 8.

Comment: I'm a little confused about the scores that were used. The methods starts by saying that the total score was used, and then that a subscore was used. I assume that the total score was the primary outcome. Then a sensitivity analysis is mentioned – I'm not clear what the difference between this and the total score is. Then the results focus on the subscore, which now seems to be the primary outcome. This all needs to be made clear and consistent throughout the methods and results.

Response: We have now clarified this in the abstract in the first sentence of the 'Outcome measures' section. We have removed the text "We linked to the 'total' score", and amended the text to state we linked to the EYFSP scores. The text in this section has been amended to state that we looked at the total and the sub-score in the analysis.

We agree that reference to sensitivity analysis confuses the presentation of the results. Both outcomes are assessed and the results of both outcomes presented in the paper. Therefore in addition to the changes noted above we have removed reference to the analysis of the total EYFSP score as sensitivity analysis.

Comment: The text states that cohort members gave their consent to access GP records – but it does not mention education records. Was explicit consent obtained for linkage with education records?

Response: Yes consent for education and health data linkage was provided by all participants (please see the patient and public involvement section on page 6).

Comment: The results section is rather brief. First sentence – please give actual numbers. Start with the full score (keep the same order as the methods and abstract) and give how many children scored above / below each cut off. Also give the numbers and percentages in each category with a diagnosis, and for each subgroup that you mention in the text. It would be helpful to include these characteristics in a table in the main text, along with the model coefficients that are currently in the supplementary material. This could be in the main text for the primary analysis (total score) and left in the appendix for the subscore.

Response: We have added the numbers and percentages suggested to the text in the results section. Also we have now included a new table (table 4) that breaks down the numbers as requested.

Comment: For the model coefficients, please provide actual p-values rather than just * <0.05 (or leave them out altogether).

Response: We have removed reference to the p-values (retaining the 95% confidence intervals) in the supplementary material document.

Comment: Supplementary material 2 seems to repeat Figure 3 from the main text.

Response: We have removed this duplication by deleting supplementary material 2.

Comment: Rather than the current Figure 4, which just presents four probabilities, it would be more helpful to present the distribution of scores for children with and without diagnoses, e.g. using histograms. This is helpful as it would show to what extent scores for the two groups of children overlap.

Response: Figure 4 does show the probability. This is the predicted result from the regression models. We agree that it would also be useful to see the distribution of EYFSP scores for children with and without an autism diagnosis. We have produced an additional figure illustrating this and included in the supplementary material (please see supplementary material 3).

Comment: Please define UPN.

Response: UPN is "Unique Pupil Number". It is a 13-character code that identifies each pupil in the local authority school system.

This definition is now included in the main text (please see the procedure section on page 11).

Reviewer 2

Comment: I have some reservations about the results as it is not particularly surprising that children with lower EYFSP scores are those that are more likely to have an autism diagnosis, given intellectual disability is associated with autism.

Response: We believe it is important to report this finding in the literature to encourage more research in this field. Multiple teachers rate children across multiple schools and assessing the predictive validity of these assessments is a worthy aspiration given these preliminary novel findings.

Comment: I am also unsure about the authors' decision to map the 'physical development - health and self-care' code to restricted repetitive behaviours as these do not seem to be measuring the same thing. Restricted repetitive behaviours are far more specific than physical development, and physical development is likely picking up on degree of disability rather than autism.

Response: Whilst we agree that they are not measuring the same thing the expert group felt that children with autism with repetitive behaviours and restrictive interests were far less likely to be able to provide self-care and so would likely have an association. This could be explored further in future research.

Comment: Please clarify numbers of children in your cohort - Tables 1 and the abstract give three different denominators.

Response: In response to point 2 we clarified in the abstract that 8,935 children were linked, 1,852 (20.7%) children scored below the cut off on the total EYFSP score and 1,355 (15.2%) children scored below the cut off on the EYFSP sub score.

This has been added to the text in the abstract, this clarifies the denominator.

Comment: Add the number of children with ASD to the text and the abstract

Response: Just under 1% (84 children) of the 8,935 children with matched GP and education records had a diagnosis of autism.

This has been added to the abstract and the text (in results section).